# A PROBABILISTIC GENERATIVE METHOD FOR SAFE PHYSICAL SYSTEM CONTROL PROBLEMS

## ABSTRACT

Controlling complex physical systems is a crucial task in science and engineering, often requiring the balance of control objectives and safety constraints. Recently, diffusion models have demonstrated a strong ability to model high-dimensional state spaces, giving them an advantage over recent deep learning and reinforcement learning-based methods in complex control tasks. However, they do not inherently address safety concerns. In contrast, while safe reinforcement learning methods consider safety, they typically fail to provide guarantees for satisfying safety constraints. To address these limitations, we propose Safe Conformal Physical system control (SafeConPhy), which optimizes the diffusion model with a provable safety bound iteratively to satisfy the safety constraint. We pre-train a diffusion model on the training set. Given the calibration set and the specific control targets, we derive a provable safety bound using conformal prediction. After iteratively enhancing the safety of the diffusion model with the progressively updated bound, the model's output can be certified as safe with a user-defined probability. We evaluate our algorithm on two control tasks: 1D Burgers' equation and 2D incompressible fluid. Our results show that our algorithm satisfies safety constraints, and outperforms prior control methods and safe offline RL algorithms.

## 1 INTRODUCTION

The control of complex physical systems is critical and essential in many scientific and engineering fields, including fluid dynamics (Hinze & Kunisch, 2001), nuclear fusion (Edwards et al., 1992), and mathematical finance (Soner, 2004). In real-world scenarios, controlling such systems often requires addressing safety concerns (Barros & des Santos, 1998; Argomedo et al., 2013). For example, in fluid dynamics, small errors in control can lead to turbulence or structural damage, while in controlled nuclear fusion, failure to maintain safety constraints could result in catastrophic consequences. Safety, in this context, involves ensuring that the control sequences guide the system to satisfy pre-defined constraints, thereby mitigating risks and preventing hazardous situations (Dawson et al., 2022; Liu et al., 2023a). Notably, safety remains a bottleneck for applying machine learning to specific scientific and engineering problems, as many machine learning algorithms lack the mechanisms to guarantee safety constraints in their control outputs. This gap between performance and safety has become a critical obstacle in deploying machine learning for high-stake applications.

Despite of its importance, the safe control of complex physical systems is challenging. Firstly, to avoid unacceptable risks, one should prevent algorithms without safety guarantees from interacting with the environment, restricting us to an offline setting with pre-collected data. However, the data is often non-optimal and may contain unsafe samples, resulting in a significant gap between the observed data distribution and the near-optimal, safe distribution (Xu et al., 2022; Liu et al., 2023a). Secondly, the algorithm must balance the need for high performance with adherence to safety constraints (Liu et al., 2023a; Zheng et al., 2024).

After many years of research on traditional control algorithms (Li et al., 2006; Protas, 2008), the advancement of neural networks leads to the emergence of numerous deep learning-based algorithms (Farahmand et al., 2017; Holl et al., 2020; Hwang et al., 2022). For complex physical systems, which are highly nonlinear and high-dimensional, the deep learning-based methods achieve outstanding results (Hwang et al., 2022; Holl et al., 2020; Wei et al., 2024). However, the above deep learning-based methods generally do not account for safety considerations. Regarding safe offline

Table 1: **Comparison between previous deep learning-based control algorithms and our proposed SafeConPhy.** SafeConPhy considers the safety constraints in the control of complex physical systems, and its safety is certifiable before interacting with the environment.

| Methods | Complex Physical System | Safety Constraint | Certifiable |
|---|:---:|:---:|:---:|
| DiffPhyCon (Wei et al., 2024) | ✓ | ✗ | ✗ |
| CDT (Liu et al., 2023b) | ✗ | ✓ | ✗ |
| TREBI (Lin et al., 2023) | ✗ | ✓ | ✗ |
| **SafeConPhy (Ours)** | ✓ | ✓ | ✓ |

reinforcement learning (RL), on the one hand, RL methods struggle to optimize long-term control sequences under the constraints of system dynamics (Wei et al., 2024). On the other hand, recent TREBI (Lin et al., 2023) and FISOR (Zheng et al., 2024) utilize diffusion models for planning and theoretically analyzing how to satisfy safety constraints, but they fail to compute the probabilistic bound of safety costs concretely. This limitation prevents their capability of certifying safety before testing, which is inconsistent with the goal of satisfying safety constraints using offline data.

To address these problems, we propose Safe Conformal Physical system control (SafeConPhy), an iterative safety improvement method with a certifiable safety bound. Firstly, in offline settings, the training data are often sub-optimal and unsafe, exhibiting a significant deviation from the desired distribution, which is optimal and safe. Inspired by concepts from conformal prediction (Vovk et al., 2005b; Tibshirani et al., 2019), we estimate the model prediction error under distribution shift based on a portion of split-out training data (called *calibration set*) and the specific control targets. With the estimated prediction error, we compute a probabilistic upper bound of the safety score, and the safety score for the model's interaction with the environment will be within the upper bound with a user-defined probability. Thus, the model's safety can be certified by verifying whether the upper bound satisfies the safety constraint. Secondly, we implement a process to improve the model safety by leveraging the upper bound through guidance and fine-tuning iteratively. The guidance step directs the model to stochastically generate multiple samples that potentially satisfy the safety constraint, while the fine-tuning step updates the model by incorporating these samples and the safety bound. This safety improvement process is iterative and continues until the safety upper bound meets the safety constraints.

In summary, the advantages of SafeConPhy are highlighted in Table 1. Our main contributions are as follows: **(1)** We introduce safety constraints into the deep learning-based control of complex physical systems, develop two datasets for safe physical system control tasks to evaluate different methods, and propose the offline algorithm SafeConPhy. **(2)** Considering the model's prediction error, we provide a certifiable upper bound of the safety score and design an iterative safety improvement process that uses the upper bound to promote the output distribution becoming more optimal and safer. **(3)** We conduct experiments on 1D Burgers' Equation and 2D incompressible fluid, whose results demonstrate that SafeConPhy can meet the safety constraints and reach better control objectives at the same time.

## 2 RELATED WORK

### 2.1 CONTROL OF PHYSICAL SYSTEMS

The development of control methods in physical systems is critical across various engineering areas, including PID (Li et al., 2006), supervised learning (SL) (Holl et al., 2020; Hwang et al., 2022), reinforcement learning (RL) (Farahmand et al., 2017; Pan et al., 2018; Rabault et al., 2019), and physics-informed neural networks (PINNs) (Mowlavi & Nabi, 2023). Among these, PID is one of the earliest and most widely used method (Johnson & Moradi, 2005), known for its simplicity and effectiveness in regulating physical systems; however, it faces challenges in parameter tuning and struggles with highly nonlinear or time-varying systems. With the advancement of deep learning, SL (Holl et al., 2020) has been applied to optimize control sequences through backpropagation over entire trajectories, but it lacks the adaptability to dynamic environments since it is typically trained on fixed datasets. To overcome the above issue, RL enhances adaptability by leveraging diverse datasets or interactions with the environment, achieving notable success in controlling physical systems, in-

cluding fluid dynamics (Novati et al., 2017; Feng et al., 2023), underwater devices (Zhang et al., 2022; Feng et al., 2024), and nuclear fusion (Degrave et al., 2022). Furthermore, the adjoint method (Protas, 2008) and PINNs (Mowlavi & Nabi, 2023) are also incorporated in PDE control, but they require an explicit form of the PDE. Currently, the diffusion model used in physical systems' control (Wei et al., 2024) integrates the learning of entire state trajectories and control sequences, enabling global optimization that incorporates the physical information learned by the model. However, it does not consider the important cases where safety constraints are required.

## 2.2 Safe Offline Reinforcement Learning

Recently, the offline setting has attracted attention in the field of safe reinforcement learning (RL), as it avoids generating dangerous behaviors through direct interaction with the environment (Achiam et al., 2017; Zhang et al., 2020; Stooke et al., 2020; Liu et al., 2022). CPQ (Xu et al., 2022) is the first practical safe offline RL method that assigns high costs to OOD and unsafe actions and updates the value function as well as the policy only with safe actions. COptiDICE (Lee et al., 2022) is a DICE-based method and corrects the stationary distribution. CDT (Liu et al., 2023b) takes the decision transformer to solve safe offline RL problems as multi-objective optimization. However, these methods lack the ability to model high-dimensional state space. More recent methods like TREBI (Lin et al., 2023) and FISOR (Zheng et al., 2024) solve it through diffusion model planning. But they do not consider the upper bound of the safety score in a probabilistic sense and differ significantly from SafeConPhy in terms of the algorithm.

## 2.3 Conformal Prediction

Conformal prediction (Vovk et al., 2005a) is a statistical framework that constructs prediction intervals guaranteed to contain the true label with a specified probability. Its validity could be compromised, however, by the violation of the core assumption of exchangeability due to distribution shifts in real-world scenarios (Chernozhukov et al., 2018; Hendrycks et al., 2018). Recent studies (Maxime Cauchois & Duchi, 2024) have extended conformal prediction to accommodate various distribution shifts. For example, Tibshirani et al. (2019) proposed weighted conformal prediction to handle covariate shift, where training and test data distributions differ. Podkopaev & Ramdas (2021) introduced reweighted conformal prediction and calibration techniques to address label shift using unlabeled target data. Adaptive conformal inference (Gibbs & Candes, 2021) provides valid prediction sets in online settings with unknown, time-varying distribution shifts without relying on exchangeability. Inspired by previous approaches, SafeConPhy establishes an upper bound on the confidence level by maintaining a weighted score set. Our method ensures the true safety value for a given control sequence lies within the bound without requiring additional assumptions about the model or data distribution, effectively addressing distribution shifts between pre-collected data and the target distribution.

## 3 Preliminary

### 3.1 Problem Setup

We consider the following safe control problem of complex physical systems:

$$\mathbf{w}^* = \arg\min_{\mathbf{w}} \mathcal{J}(\mathbf{u}, \mathbf{w}) \quad \text{s.t.} \quad \mathcal{C}(\mathbf{u}, \mathbf{w}) = 0, \quad s(\mathbf{u}) \leq s_0, \tag{1}$$

where $\mathbf{u}(t, \mathbf{x}) : [0, T] \times \Omega \mapsto \mathbb{R}^{d_{\mathbf{u}}}$ is the system's state trajectory with dimension $d_{\mathbf{u}}$ and $\mathbf{w}(t, \mathbf{x}) : [0, T] \times \Omega \mapsto \mathbb{R}^{d_{\mathbf{w}}}$ is the external control signal with dimension $d_{\mathbf{w}}$. They are both defined on the time range $[0, T] \subset \mathbb{R}$ and spatial domain $\Omega \subset \mathbb{R}^D$. $\mathcal{J}(\mathbf{u}, \mathbf{w})$ is the objective of the control problem, and $\mathcal{C}(\mathbf{u}, \mathbf{w}) = 0$ is the physical constraint, such as the partial differential equation. As for the safety constraint, $s(\mathbf{u})$ is the safety score and $s_0$ is the bound of the safety score. We need to minimize the control objective while satisfying physical constraints and constraining the safety score to stay below the bound, which requires a careful balance between safety and performance. However, it is important to note that safety and performance are not on equal footing, and the pursuit of a better objective should be built upon ensuring safety.

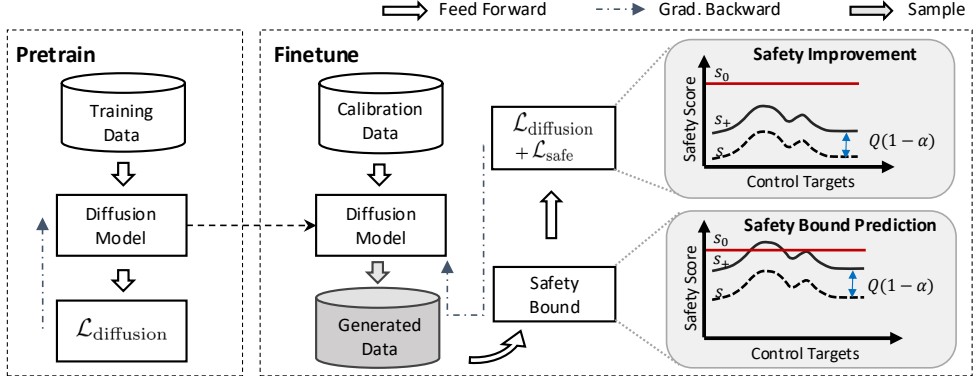

Figure 1: **Overview of SafeConPhy**. First, we pre-train a diffusion model $p_\theta$ on the training data. Then, we derive a safety bound to certify that the model satisfies the safety constraints. To satisfy the safety constraint, we further design a loss function term based on the safety bound to optimize the diffusion model.

## 3.2 DIFFUSION MODELS AND DIFFUSION CONTROL

Diffusion models (Ho et al., 2020) learn data distribution from data in a generative way. They present impressive performance in a broad range of generation tasks. Diffusion models involve diffusion/denoising processes: the diffusion process $q(\mathbf{x}^{k+1}|\mathbf{x}^k) = \mathcal{N}(\mathbf{x}^{k+1}; \sqrt{\alpha_k}\mathbf{x}_k, (1-\alpha_k)\mathbf{I})$ corrupts the data distribution $p(\mathbf{x}_0)$ to a prior distribution $\mathcal{N}(\mathbf{0}, \mathbf{I})$, and the denoising process $p_\theta(\mathbf{x}^{k-1}|\mathbf{x}^k) = \mathcal{N}(\mathbf{x}^{k-1}; \mu_\theta(\mathbf{x}^k, k), \sigma_k\mathbf{I})$ makes sampling in a reverse direction. Here $k$ is the diffusion/denoising step, $\{\alpha_k\}_{k=1}^K$ and $\{\sigma_k\}_{k=1}^K$ are the noise and variance schedules. In practice, a denoising network $\boldsymbol{\epsilon}_\theta$ is trained to estimate the noise to be removed in each step. During inference, the iterative application of $\boldsymbol{\epsilon}_\theta$ from the prior distribution could generate a new sample that approximately follows the data distribution $p(\mathbf{x})$.

Recently, DiffPhyCon (Wei et al., 2024) applies diffusion models to solve the control problem as in Eq. 1 without the safety constraint $s(\mathbf{u}) \leq s_0$. For brevity, we only summarize its light version. It transforms the physical constraint to a parameterized energy-based model (EBM) $E_\theta(\mathbf{u}, \mathbf{w})$ with the correspondence $p(\mathbf{u}, \mathbf{w}) \propto \exp(-E_\theta(\mathbf{u}, \mathbf{w}))$. Then the problem is converted to an unconstrained optimization over $\mathbf{u}$ and $\mathbf{w}$ for all physical time steps simultaneously:

$$\mathbf{u}^*, \mathbf{w}^* = \arg\min_{\mathbf{u}, \mathbf{w}} \left[ E_\theta(\mathbf{u}, \mathbf{w}) + \lambda \cdot \mathcal{J}(\mathbf{u}, \mathbf{w}) \right], \tag{2}$$

where $\lambda$ is a hyperparameter. To optimize $E_\theta$, a denoising network $\boldsymbol{\epsilon}_\theta$ is trained to approximate $\nabla_{\mathbf{u}, \mathbf{w}} E_\theta(\mathbf{u}, \mathbf{w})$ by the following loss:

$$\mathcal{L} = \mathbb{E}_{k \sim U(1,K), (\mathbf{u}, \mathbf{w}) \sim p(\mathbf{u}, \mathbf{w}), \boldsymbol{\epsilon} \sim \mathcal{N}(\mathbf{0}, \mathbf{I})}[\|\boldsymbol{\epsilon} - \boldsymbol{\epsilon}_\theta(\sqrt{\bar{\alpha}_k}[\mathbf{u}, \mathbf{w}] + \sqrt{1 - \bar{\alpha}_k}\boldsymbol{\epsilon}, k)\|_2^2], \tag{3}$$

where $\bar{\alpha}_k := \prod_{i=1}^k \alpha_i$. After $\boldsymbol{\epsilon}_\theta$ is trained, Eq. 2 can be optimized by sampling from an initial sample $(\mathbf{u}^K, \mathbf{w}^K) \sim \mathcal{N}(\mathbf{0}, \mathbf{I})$, and iteratively running the following process

$$(\mathbf{u}^{k-1}, \mathbf{w}^{k-1}) = (\mathbf{u}^k, \mathbf{w}^k) - \eta\left(\boldsymbol{\epsilon}_\theta([\mathbf{u}^k, \mathbf{w}^k], k) + \lambda\nabla_{\mathbf{u}, \mathbf{w}}\mathcal{G}(\hat{\mathbf{u}}^k, \hat{\mathbf{w}}^k) + \xi, \quad \xi \sim \mathcal{N}(\mathbf{0}, \sigma_k^2\mathbf{I}) \tag{4}$$

under the guidance of $\mathcal{G} = \mathcal{J}$ for $k = K, K-1, ..., 1$. Here $[\hat{\mathbf{u}}^k, \hat{\mathbf{w}}^k]$ is the noise-free estimation of $[\mathbf{u}^0, \mathbf{w}^0]$. The final sampling step yields the solution $\mathbf{w}^0$ for the optimization problem in Eq. 2.

## 4 METHOD

In this section, we introduce our proposed method SafeConPhy, with its overall framework outlined in Figure 1. First, in Section 4.1, we briefly outline the overall steps of the algorithm. Next, in Section 4.2, we explain how conformal prediction is applied to estimate the safety score $s$ under distribution shift in a probabilistic sense, and theoretically derive the formula for the provable safety bound $s_+$. Finally, in Section 4.3, we detail the implementation of the entire algorithm. Specifically,

---

**Algorithm 1** Inference of SafeConPhy

---

1: **Require** Calibration set $D_{\text{cal}}$, training set $D_{\text{train}}$, confidence level $\alpha$, control objective $\mathcal{J}(\cdot)$, safety score $s(\cdot)$, number of iterations $N$
2: **for** $n = 1, \ldots, N$ **do**
3:     Compute the weighted score set $\tilde{\mathcal{S}}$ with $D_{\text{cal}}$ // Eq. 11
4:     Get the quantile $Q(1 - \alpha; \tilde{\mathcal{S}})$
5:     Sample the control sequence $\mathbf{w}$ with guidance $\mathcal{G}$ // Eq. 14
6:     Compute $s_+(\tilde{\mathbf{u}}_\theta(\mathbf{w}))$ with conditionally sampled $\tilde{\mathbf{u}}_\theta(\mathbf{w})$ // Eq. 12
7:     Take gradient descent step on $\nabla_\theta \mathcal{L}_{\text{fine-tune}}$ // Eq. 15
8: **end for**
9: Sample the control sequence $\mathbf{w}$ with guidance $\mathcal{G}$ // Eq. 14
10: **return** $\mathbf{w}$

---

we first describe the detailed implementation of this estimation. Additionally, we describe how the estimated safety score is utilized to modify the distribution of generated control sequences through guidance and fine-tuning.

## 4.1 OVERALL PROCEDURES

In this section, we present the workflow of our algorithm as in Figure 1. We set aside a portion of the original training data as the *calibration set* $D_{\text{cal}}$, which will be used later to estimate the model's prediction errors. The remaining data, which will be used for actual training, is referred to as the training data $D_{\text{train}}$. After pre-training with $D_{\text{train}}$ as described in Eq. 3, we get the diffusion model $p_\theta$ which models the joint distribution of $[\mathbf{u}, \mathbf{w}]$.

Next, we conduct Iterative Safety Improvement, where we apply the provable safety bound $s_+$ under distribution shift to enhance model safety iteratively. First, the calculation of the safety bound $s_+$ runs throughout the entire loop. The safety bound basically takes the calibration set to obtain a corresponding set of model prediction errors (called the *score set*). Furthermore, we take into account the distribution shift between the calibration set and data generated based on control targets, so we apply weighting to the set of model prediction errors to get the *weighted score set* as Eq. 11. Then, we use the quantile of this set to represent the error in the model's predicted safety score.

Second, 'iterative' refers to the process where we cyclically use the guidance to generate samples, and then combine these samples with model prediction errors to fine-tune the model parameters, thereby improving the model's safety. Specifically, in each iteration, we sample the control sequence $\mathbf{w}$ under the guidance as described in Eq. 14 containing $s_+$. After getting $\mathbf{w}$, we conditionally sample $\tilde{\mathbf{u}}_\theta(\mathbf{w})$ to get the provable safety bound $s_+(\tilde{\mathbf{u}}_\theta(\mathbf{w}))$. And we can now take the fine-tuning loss $\mathcal{L}_{\text{fine-tune}}$ involving the training data $D_{\text{train}}$ and also the progressively updated $s_+(\tilde{\mathbf{u}}_\theta(\mathbf{w}))$ as in Eq. 15 to fine-tune the model parameters $\theta$.

Finally, after several iterations, we use the fine-tuned diffusion model, once again under the influence of guidance, to generate the control sequences, which serve as the final output of the algorithm. The complete inference process can be seen in Algorithm 1.

## 4.2 PROVABLE SAFETY BOUND WITH CONFORMAL PREDICTION

In offline safety control problems, the gap between pre-collected data and the target distribution exacerbates models' prediction errors, which can be critical in ensuring safety. To address this issue, we employ the conformal prediction technique to obtain a provable safety bound, ensuring that the true safety score is included within this estimate with a provable level of confidence, without requiring additional assumptions about the model or the data distribution.

**Calibration set and score set.** To achieve the goal mentioned above, we first set aside a portion of the training dataset, which is not used for training, as the *calibration set* $D_{\text{cal}}$. After training, we take the calibration set to obtain the *score set*, which is defined as

$$\mathcal{S} \coloneqq \{|s(\tilde{\mathbf{u}}_\theta(\mathbf{w}_i)) - s(\mathbf{u}_i)| : (\mathbf{u}_i, \mathbf{w}_i) \in D_{\text{cal}}\}. \tag{5}$$

Here $\mathbf{u}_i$ and $\mathbf{w}_i$ represent different samples, $\tilde{\mathbf{u}}_\theta(\mathbf{w})$ is the system state conditioned on control $\mathbf{w}_i$ and predicted by the model, and $\theta$ is the parameters of the model. The score set can be considered as recording the estimation errors of the model with respect to the safety score $s$.

**Weighted score set under distribution shift.** However, since the data distribution in the calibration set differs from the final model-generated data distribution used for control, it does not satisfy the exchangeability[1] condition required by conformal prediction (Vovk et al., 2005a; Papadopoulos et al., 2002; Lei et al., 2016). Intuitively, each sample in the calibration set has a different probability of appearing in the final data distribution generated by the model, so we further apply weighting to the elements of the score set.

We define $p(\mathbf{u}, \mathbf{w})$ as the distribution of the calibration set. And we let $\tilde{p}(\mathbf{u}, \mathbf{w})$ be the distribution of the test data, where $\mathbf{w}$ is generated by the model based on the control task and safety constraints, and $\mathbf{u}$ is the true system state obtained from the interaction between the control sequence $\mathbf{w}$ and the environment. According to conformal prediction under covariate shift (Tibshirani et al., 2019), the calculation of the weights is as follows:

$$\omega(\mathbf{u}_i, \mathbf{w}_i) := \frac{\mathrm{d}\tilde{p}(\mathbf{u}_i, \mathbf{w}_i)}{\mathrm{d}p(\mathbf{u}_i, \mathbf{w}_i)} = \frac{\mathrm{d}\tilde{p}(\mathbf{w}_i)\mathrm{d}\tilde{p}(\mathbf{u}_i|\mathbf{w}_i)}{\mathrm{d}p(\mathbf{w}_i)\mathrm{d}p(\mathbf{u}_i|\mathbf{w}_i)}, \tag{6}$$

where $\mathrm{d}p$ denotes the probability density function of distribution $p$. Since $\mathrm{d}\tilde{p}(\mathbf{u}_i|\mathbf{w}_i)$ and $\mathrm{d}p(\mathbf{u}_i|\mathbf{w}_i)$ both represent the physical constraints of the system itself, the weights can be simplified as

$$\omega(\mathbf{u}_i, \mathbf{w}_i) = \frac{\mathrm{d}\tilde{p}(\mathbf{w}_i)}{\mathrm{d}p(\mathbf{w}_i)}. \tag{7}$$

We note that as described in Eq. 14, $\mathbf{w}_i$ is generated by the energy-based model under guidance $\mathcal{G}$. In the safe control problems, guidance $\mathcal{G}$ encompasses both the control objective and safety constraints and will be detailed in Section 4.3. With the influence of $\mathcal{G}$, $\tilde{p}(\mathbf{u}_i, \mathbf{w}_i) \propto \exp(-E_\theta(\mathbf{u}_i, \mathbf{w}_i) - \mathcal{G}(\mathbf{u}_i, \mathbf{w}_i)) \propto p_\theta(\mathbf{u}_i, \mathbf{w}_i)\exp(-\mathcal{G}(\mathbf{u}_i, \mathbf{w}_i))$, where $p_\theta$ is distribution learned by the diffusion model. Thus we obtain

$$\omega(\mathbf{u}_i, \mathbf{w}_i) = C\frac{\mathrm{d}p_\theta(\mathbf{u}_i, \mathbf{w}_i)e^{-\mathcal{G}(\mathbf{u}_i, \mathbf{w}_i)}}{\mathrm{d}p(\mathbf{u}_i, \mathbf{w}_i)}, \tag{8}$$

where $C$ is a constant. Given that the calibration set and the training dataset follow the same distribution, and assuming that the impact of the diffusion model's learning error on the dataset is sufficiently small relative to the second term $e^{-\mathcal{G}(\mathbf{u}_i, \mathbf{w}_i)}$, we can approximate the weight as

$$\omega(\mathbf{u}_i, \mathbf{w}_i) = Ce^{-\mathcal{G}(\mathbf{u}_i, \mathbf{w}_i)}. \tag{9}$$

Finally, we normalize the weight and obtain

$$\hat{\omega}(\mathbf{u}_i, \mathbf{w}_i) = \frac{Ce^{-\mathcal{G}(\mathbf{u}_i, \mathbf{w}_i)}}{\sum_{(\mathbf{u}_i, \mathbf{w}_i) \in D_{\mathrm{cal}}} Ce^{-\mathcal{G}(\mathbf{u}_j, \mathbf{w}_j)}} = \frac{e^{-\mathcal{G}(\mathbf{u}_i, \mathbf{w}_i)}}{\sum_{(\mathbf{u}_i, \mathbf{w}_i) \in D_{\mathrm{cal}}} e^{-\mathcal{G}(\mathbf{u}_j, \mathbf{w}_j)}}. \tag{10}$$

The *weighted score set* is then defined as

$$\tilde{\mathcal{S}} := \{\hat{\omega}(\mathbf{u}_i, \mathbf{w}_i)|s(\tilde{\mathbf{u}}_\theta(\mathbf{w}_i)) - s(\mathbf{w}_i)| : (\mathbf{u}_i, \mathbf{w}_i) \in D_{\mathrm{cal}}\}. \tag{11}$$

**Upper bound $s_+$ on the confidence level $\alpha$.** For a given control sequence $\mathbf{w}$, we provide an upper bound $s_+$ below, such that with at least $1 - \alpha$ probability, the true $s$ is smaller than $s_+$. In detail, we exploit the weighted score set to define the $s_+$ as

$$s_+(\tilde{\mathbf{u}}_\theta(\mathbf{w})) = s(\tilde{\mathbf{u}}_\theta(\mathbf{w})) + Q(1 - \alpha; \tilde{\mathcal{S}}), \tag{12}$$

where $Q(1-\alpha; \tilde{\mathcal{S}})$ is $\mathrm{Quantile}((1-\alpha)(1 + \frac{1}{|D_{\mathrm{cal}}|}); \tilde{\mathcal{S}})$ and is still a differentiable function with respect to $\theta$, and $|D_{\mathrm{cal}}|$ is the cardinality of $D_{\mathrm{cal}}$. The precise and formal meaning of the probabilistic upper bound is demonstrated through the following lemma.

**Lemma 1.** *Assume samples $(\mathbf{u}_i, \mathbf{w}_i) \sim p$ in the calibration set are independent, and the test set $(\mathbf{u}, \mathbf{w}) \sim \tilde{p}$ is also independent with the calibration set. Assume $p$ is absolutely continuous with respect to $\tilde{p}$, $s_+$ is defined as in Eq. 12, then*

$$\mathbb{P}[s(\mathbf{u}(\mathbf{w})) \leq s_+(\tilde{\mathbf{u}}_\theta(\mathbf{w}))] \geq 1 - \alpha, \tag{13}$$

*where $\mathbf{u}(\mathbf{w})$ is the real system state corresponding to the control sequence $\mathbf{w}$, and $\tilde{\mathbf{u}}_\theta(\mathbf{w})$ is the system state predicted by the model conditioned on the same control.*

---

[1]$(\mathbf{w}_i, s_i)_{i=1}^N$ are exchangeable if, for any permutation $\sigma$ of $[\![1, N]\!]$, $\mathcal{P}((\mathbf{u}_1, s_1), \cdots, (\mathbf{u}_N, s_N)) = \mathcal{P}((\mathbf{u}_{\sigma(1)}, s_{\sigma(1)}), \cdots, (\mathbf{u}_{\sigma(N)}, s_{\sigma(N)}))$, where $\mathcal{P}$ is the joint distribution.

### 4.3 TARGET CONTROL GENERATION BASED ON ESTIMATED SAFETY

Next, based on the deduced $s_+$, we describe the detailed implementation of modules in SafeConPhy.

**Conditionally sample $\tilde{\mathbf{u}}_\theta(\mathbf{w})$.** In our proposed algorithm, we need to sample from the conditional distribution $p(\mathbf{u}|\mathbf{w})$ with the model that learns the joint distribution $p(\mathbf{u}, \mathbf{w})$. To achieve this, at each denoising step of the sampling process, we replace the noisy $\mathbf{w}$ in the input of the denoising network with the actual clean $\mathbf{w}$ that serves as the condition (Chung et al., 2023). In fact, this situation represents a special case of the data distribution that the denoising network encounters during training, where the $\mathbf{u}$ part is noisy, while the $\mathbf{w}$ part remains noise-free.

**Guidance $\mathcal{G}$.** Guidance is the first method we adopt to steer the model's output toward satisfying both the control objectives and safety constraints. It plays a role during the sampling process of the diffusion model. When considering safety, the specific form of our guidance is as follows:

$$\mathcal{G}(\mathbf{u}, \mathbf{w}) = \mathcal{J}(\mathbf{u}, \mathbf{w}) + \gamma \max[s_+(\mathbf{u}(\mathbf{w})) - s_0, 0]. \tag{14}$$

Note that although we follow the previous symbol $s_+(\mathbf{u}(\mathbf{w}))$, during sampling, $\mathbf{u}$ and $\mathbf{w}$ are actually generated by the diffusion model jointly but not conditionally. The specific denoising step of implementing guidance follows Eq. 4.

**Fine-tuning.** The second method for adjusting the output data distribution of the model is fine-tuning, which achieves the adjustment by optimizing the model parameters $\theta$. Specifically, both terms in $s_+$, as shown in Eq. 12, are functions of $\theta$. Therefore, it is both reasonable and effective to compute the gradient of $s_+$ with respect to $\theta$ to take the gradient descent step. It is worth noting that retaining the computation graph for all denoising steps with respect to $\theta$ would result in an unmanageable memory overhead. Therefore, when we need to keep the computation graph during denoising for gradient calculation, we only retain the computation graph of the final denoising step.

To optimize the safety score and the diffusion loss (referred to Eq. 3) simultaneously, we form the fine-tune loss $\mathcal{L}_{\text{fine-tune}}$ as the weighted sum of both the safety loss $\mathcal{L}_{\text{safe}}$ and diffusion loss $\mathcal{L}_{\text{diffusion}}$:

$$\begin{aligned}
\mathcal{L}_{\text{fine-tune}} &= \mathcal{L}_{\text{safe}} + \beta \mathcal{L}_{\text{diffusion}} \\
&= \sum_{\mathbf{w} \in D_{\text{sampled}}} \max[s_+(\tilde{\mathbf{u}}_\theta(\mathbf{w})) - s_0, 0] \\
&+ \beta \sum_{(\mathbf{u}, \mathbf{w}) \in D_{\text{train}}} \|\boldsymbol{\epsilon} - \boldsymbol{\epsilon}_\theta(\sqrt{\bar{\alpha}_k}[\mathbf{u}, \mathbf{w}] + \sqrt{1 - \bar{\alpha}_k}\boldsymbol{\epsilon}, \mathbf{u}_0, k)\|_2^2,
\end{aligned} \tag{15}$$

where $\mathbf{w}$ in the first term is from $D_{\text{sampled}}$ sampled according to the guidance described above, $(\mathbf{u}, \mathbf{w})$ in the second term is from the training set $D_{\text{train}}$, $k$ is the denoising step, $\boldsymbol{\epsilon} \sim \mathcal{N}(0, \mathbf{I})$, $\bar{\alpha}_k := \prod_{i=1}^k \alpha_i$ is the product of noise schedules and $[\mathbf{u}, \mathbf{w}]$ means the concatenation of $\mathbf{u}$ and $\mathbf{w}$.

## 5 EXPERIMENT

To verify our statements that SafeConPhy can both achieve safety and reach lower control objectives than other methods, we conduct experiments on safe offline control problems on 1D Burgers' equation and 2D incompressible fluid. Besides, to evaluate the quality of safety, for different problems, we provide several corresponding metrics.

For comparison, we choose imitation learning method Behavior Cloning (*BC*) (Pomerleau, 1988), and safe reinforcement learning and imitation learning methods involving *BC* with safe data filtering (*BC-Safe*), Constrained Decision Transformer (*CDT*) (Liu et al., 2023b) and *CDT* with safe data filtering (*CDT-Safe*), diffusion-based method *TREBI* (Lin et al., 2023). Note that *CDT* shows the best performance in the Offline Safe RL benchmark OSRL (Liu et al., 2023a). In addition, we combine the physical system control method Supervised Learning (Hwang et al., 2022) with the Lagrangian approach (Chow et al., 2018) (*SL-Lag*) to enforce safety constraints. We also apply the classical control method *PID* (Li et al., 2006). We provide the anonymous code here.

### 5.1 1D BURGERS' EQUATION

**Experiment settings.** 1D Burgers' equation is a fundamental equation that governs various physical systems including fluid dynamics and gas dynamics. Here we follow previous works (Hwang et al.,

Table 2: **Results of 1D Burgers' equation.** Gray: $s_{\text{norm}}$ is greater than 1 (unsafe). Black: $s_{\text{norm}}$ is smaller than 1 (safe). **Bold**: Safe trajectories with *lowest* $\mathcal{J}$.

| Methods | $\mathcal{J} \downarrow$ | $s_{\text{norm}} \downarrow$ | $\mathcal{R}_{\text{sample}} \downarrow$ | $\mathcal{R}_{\text{time}} \downarrow$ | $\mathcal{R}_{\text{point}} \downarrow$ |
|---|---|---|---|---|---|
| BC | 0.0001 | 1.9954 | 38% | 13% | 1.2% |
| BC-Safe | 0.0002 | 1.9601 | 14% | 3% | 0.2% |
| PID | 0.0968 | 0.5691 | 0% | 0% | 0.0% |
| SL-Lag | 0.0115 | 0.6817 | 0% | 0% | 0.0% |
| CDT | 0.0026 | 1.9220 | 8% | 1% | 0.1% |
| CDT-Safe | 0.0021 | 0.8570 | 0% | 0% | 0.0% |
| TREBI | 0.0074 | 0.7821 | 0% | 0% | 0.0% |
| **SafeConPhy (Ours)** | **0.0011** | **0.8388** | **0%** | **0%** | **0.0%** |

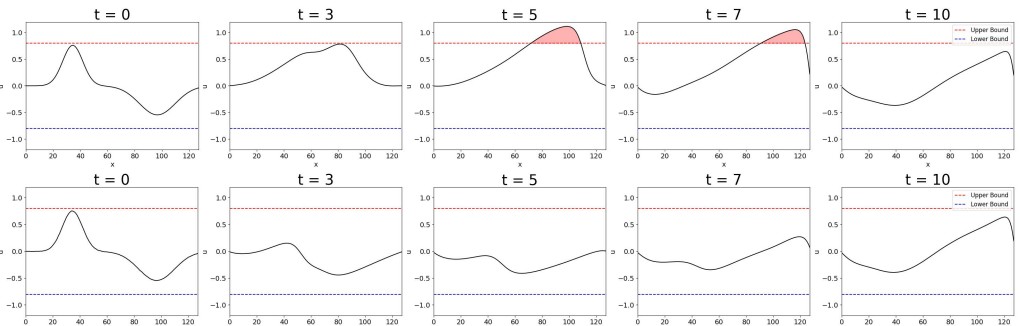

Figure 2: **Visualizations of the 1D Burgers' equation.** The top row shows the original trajectory corresponding to the control target, and the bottom row is the trajectory controlled by SafeConPhy.

2022; Mowlavi & Nabi, 2023) and consider the Dirichlet boundary condition along with an external force $\mathbf{w}(t,x)$. This equation is formulated as follows:

$$\begin{cases} \frac{\partial \mathbf{u}(t,x)}{\partial t} = -\mathbf{u}(t,x) \cdot \frac{\partial \mathbf{u}(t,x)}{\partial x} + \nu \frac{\partial^2 \mathbf{u}(t,x)}{\partial x^2} + \mathbf{w}(t,x) & \text{in } [0,T] \times \Omega \\ \mathbf{u}(t,x) = 0 & \text{on } [0,T] \times \partial\Omega \\ \mathbf{u}(0,x) = \mathbf{u}_0(x) & \text{in } \{t=0\} \times \Omega, \end{cases} \tag{16}$$

where $\nu$ denotes the viscosity parameter, while $\mathbf{u}_0$ signifies the initial condition. We set $\nu = 0.01$, $T = 1$ and $\Omega = [0,1]$. Given a target state $\mathbf{u}_d(x)$, the primary control objective $\mathcal{J}$ is to minimize the control error between the final state $\mathbf{u}_T$ and the target state $\mathbf{u}_d$.

$$\mathcal{J} := \int_\Omega |\mathbf{u}(T,x) - \mathbf{u}_d(x)|^2 \mathrm{d}x. \tag{17}$$

Considering the safety constraint $s_0$, the safety score is defined as:

$$s(\mathbf{u}) := \sup_{(t,x) \in [0,T] \times \Omega} \{\mathbf{u}(t,x)^2\}. \tag{18}$$

If $s(\mathbf{u}) > s_0$, the state trajectory $\mathbf{u}$ is unsafe, and if $s(\mathbf{u}) \leq s_0$, the state trajectory $\mathbf{u}$ is safe. The bound of safety score $s_0$ is set to 0.64 in our experiment. According to this bound, 89.7% of samples are unsafe among the training set, 90% of samples are unsafe among the calibration set and all of the samples in the test set are unsafe. More details can be found in Appendix C.1.

To better evaluate whether the set of state trajectories controlled by the model is safe, we define the normalized safety score:

$$s_{\text{norm}} := \frac{1}{|\mathcal{N}_1|} \sum_{i \in \mathcal{N}_1} \frac{s(\mathbf{u}_i)}{s_0} + \frac{1}{|\mathcal{N}_2|} \sum_{i \in \mathcal{N}_2} \frac{s(\mathbf{u}_i)}{s_0}, \tag{19}$$

where $\mathcal{N}_1 = \{i \mid \mathbf{u}_i \leq s_0\}$, $\mathcal{N}_2 = \{i \mid \mathbf{u}_i > s_0\}$. Note that $s(\mathbf{u})$ and $s_0$ are always non-negative. If the state trajectories are all safe, the score is smaller than 1; If *any* state trajectory is unsafe, the cost



Figure 3: **Visualization of the 2D incompressible fluid control problem.**

is greater than 1. Therefore, when $s_{norm}$ is less than 1, the different algorithms only need to compare the control objective $\mathcal{J}$.

Additionally, we compute three unsafe rates to assess the safety levels of different methods' control results. $\mathcal{R}_{sample}$ denotes the proportion of unsafe trajectories among total trajectories[2]; $\mathcal{R}_{time}$ denotes the proportion of unsafe timesteps among all timesteps; $\mathcal{R}_{point}$ denotes the proportion of unsafe spatial lattice points in all spatial lattice points across all time steps.

**Results.** In Table 2, We report the results of the control objective $\mathcal{J}$, the safety score $s_{norm}$ and other safe metrics of different methods. SafeConPhy can meet the safety constraint and achieve the best control objective at the same time. As shown in Figure 2, given the initial condition and the final state (control target), SafeConPhy can control a state trajectory that satisfies the safety constraint and control target. Other methods either suffer from constraint violations or suboptimal objectives. *BC* and *BC-Safe* trained from expert trajectories failed to meet the safety constraints, showing that simple behavior cloning is not feasible in this control task. *SL-Lag* attempts to use the Lagrangian method to balance the control objective and safety, but this coupled training program makes it difficult to find the right balance, with a poor control error. *CDT* uses the complex Transformer architecture, which can achieve low control error, but it needs to filter unsafe data (*CDT-Safe*) to meet safety constraints. The diffusion-based method *TREBI* sacrifices too much control error to satisfy the safety constraints, because its error bound is soft.

### 5.2 2D INCOMPRESSIBLE FLUID

**Experiment settings.** We then consider the control problems of 2D incompressible fluid, which follows the Navier-Stokes equation:

$$\begin{cases} \frac{\partial \mathbf{v}}{\partial t} + \mathbf{v} \cdot \nabla \mathbf{v} - \nu \nabla^2 \mathbf{v} + \nabla p = f, \\ \nabla \cdot \mathbf{v} = 0, \\ \mathbf{v}(0, \mathbf{x}) = \mathbf{v}_0(\mathbf{x}), \end{cases} \quad (20)$$

where $\mathbf{v}$ is the velocity, $p$ is the pressure, $f$ is the external force and $\nu$ is the viscosity coefficient.

Following previous works (Holl et al., 2020; Wei et al., 2024), the control task we consider is to maximize the amount of smoke that passes through the target bucket in the fluid flow with obstacles and openings, while constraining the amount of smoke passing through the dangerous region under the safety bound. Specifically, referring to Figure 3, the control objective $\mathcal{J}$ is defined as the negative rate of smoke passing through the target bucket located at the center top, while the safety score $s$ corresponds to the rate of smoke entering the hazardous red region. It is important to note that there is a trade-off between controlling the flow through the hazardous region and achieving a more optimal control objective, which imposes higher demands on the algorithm. We set the safety score bound to $s_0 = 0.1$.

Moreover, this control task is particularly challenging due to its specific setup: not only does it require indirect control, which means that control can only be applied to the peripheral region, but the spatial control parameters reach as many as 1,792. As for safety, among all the training data, 53.1% of the samples are unsafe, meaning their safety score $s$ exceeds the bound $s_0 = 0.1$. The average safety score of the dataset is 0.3215. Other details can be found in Appendix D.1.

**Results.** We report results of SafeConPhy and baselines in Table 3. Here PID is inapplicable and SL-Lag fails to achieve reasonable control results. Due to the challenges of the task, no method can guarantee that all samples meet the safety requirements, so we introduce additional metrics to assess

---

[2]If any point in the full trajectory is unsafe, this trajectory is unsafe. So $\mathcal{R}_{sample}$ is the most stringent metric.

Table 3: **2D incompressible fluid control results.** Gray: $s_{\text{norm}}$ is greater than 1 (unsafe). Black: $s_{\text{norm}}$ is smaller than 1 (safe). **Bold**: methods marked in black with lowest $\mathcal{J}$.

| Methods | $\mathcal{J} \downarrow$ | $s_{\text{norm}} \downarrow$ | $\max[s - s_0, 0] \downarrow$ | $\mathcal{R} \downarrow$ |
|---|---|---|---|---|
| BC | -0.7125 | 7.3402 | 0.7160 | 88% |
| BC-Safe | -0.2520 | 0.3463 | 0.0330 | 8% |
| CDT | -0.7133 | 3.0778 | 0.2726 | 34% |
| CDT-Safe | -0.6360 | 0.5073 | 0.0292 | 18% |
| TREBI | -0.6105 | 0.9096 | 0.0537 | 30% |
| **SafeConPhy (Ours)** | **-0.7035** | **0.6092** | **0.0380** | **14%** |

safety. Here we define the normalized safety score $s_{\text{norm}}$ as $s/s_0$ and define $\mathcal{R}$ as the rate of unsafe samples. Additionally, we introduce another metric $\max[s - s_0, 0]$. When $s$ does not exceed the bound $s_0$, this metric is 0. If $s$ exceeds $s_0$, the metric reflects the amount by which it is surpassed. From the results, we can see that our method successfully keeps $s_{\text{norm}}$ below 1, and other safety metrics are also comparable to other baselines marked in black ($s_{\text{norm}} \leq 1$). Additionally, among the methods highlighted in black, SafeConPhy achieves a much lower $\mathcal{J}$ than others, even reaching control performance similar to methods that do not consider safety like BC.

## 5.3 Ablation Study

Table 4: **Results of the ablation study.** We compare SafeConPhy with SafeConPhy w/o fine-tuning.

| | 1D | | 2D | |
|---|---|---|---|---|
| | SafeConPhy | w/o fine-tuning | SafeConPhy | w/o fine-tuning |
| $\mathcal{J} \downarrow$ | 0.0011 | 0.0006 | -0.7035 | -0.6105 |
| $s_{\text{norm}} \downarrow$ | 0.8388 | 2.3743 | 0.6092 | 0.9096 |

We highlight that one key distinction between our framework and previous safe RL methods lies in the introduction of fine-tuning within Iterative Safety Improvement, which updates the model parameters based on specific control tasks and safety constraints. To further validate the effectiveness of our proposed Iterative Safety Improvement, we conduct experiments using a version of SafeConPhy without the fine-tuning component. As shown in the table, without fine-tuning, SafeConPhy exhibits a significant decline in safety, both in 1D and 2D settings, with the 1D case becoming notably unsafe. This emphasizes the importance and effectiveness of the Iterative Safety Improvement framework in addressing safety control problems.

## 6 Limitation and Future Work

Firstly, both experiments presented in the paper are not real-world experiments. However, our method is not constrained by specific scenarios, meaning it can be applied to real-world tasks as well, which is our future work. Secondly, we consider extending this method to other generative methods that require constraints, not just the diffusion model. Finally, in the future, we will explore the possibility of developing a stricter bound that sacrifices less accuracy while still ensuring safety.

## 7 Conclusion

In this paper, we have introduced Safe Conformal Physical system control (SafeConPhy), a probabilistic generative method for safe control problems of complex physical systems. Targeting the meaningful and important offline setting, we provide a provable probabilistic estimate of the safety score's upper bound. We then perform guidance and finetuning with this provable safety bound iteratively, improving the safety and certifying it with a user-defined probability. Experiment results on 1D Burgers' equation and 2D incompressible fluid demonstrate that on the basis of satisfying safety constraints, SafeConPhy is able to achieve a lower control objective. We believe that our method is beneficial for making machine learning-based physical control safer, improving the trustworthiness for deploying to the real world.

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

# A   VISUALIZATION OF EXPERIMENT RESULTS

## A.1   1D BURGERS' EQUATION

In this section, we provide additional visualizations of the control results for the 1D Burgers' equation, as shown in Figure 4. In these figures, the top row represents the original trajectories corresponding to the control targets, while the bottom row displays the trajectories controlled by SafeConPhy. It can be observed that SafeConPhy successfully controls the trajectories, preventing boundary violations and guiding them to the desired final state.

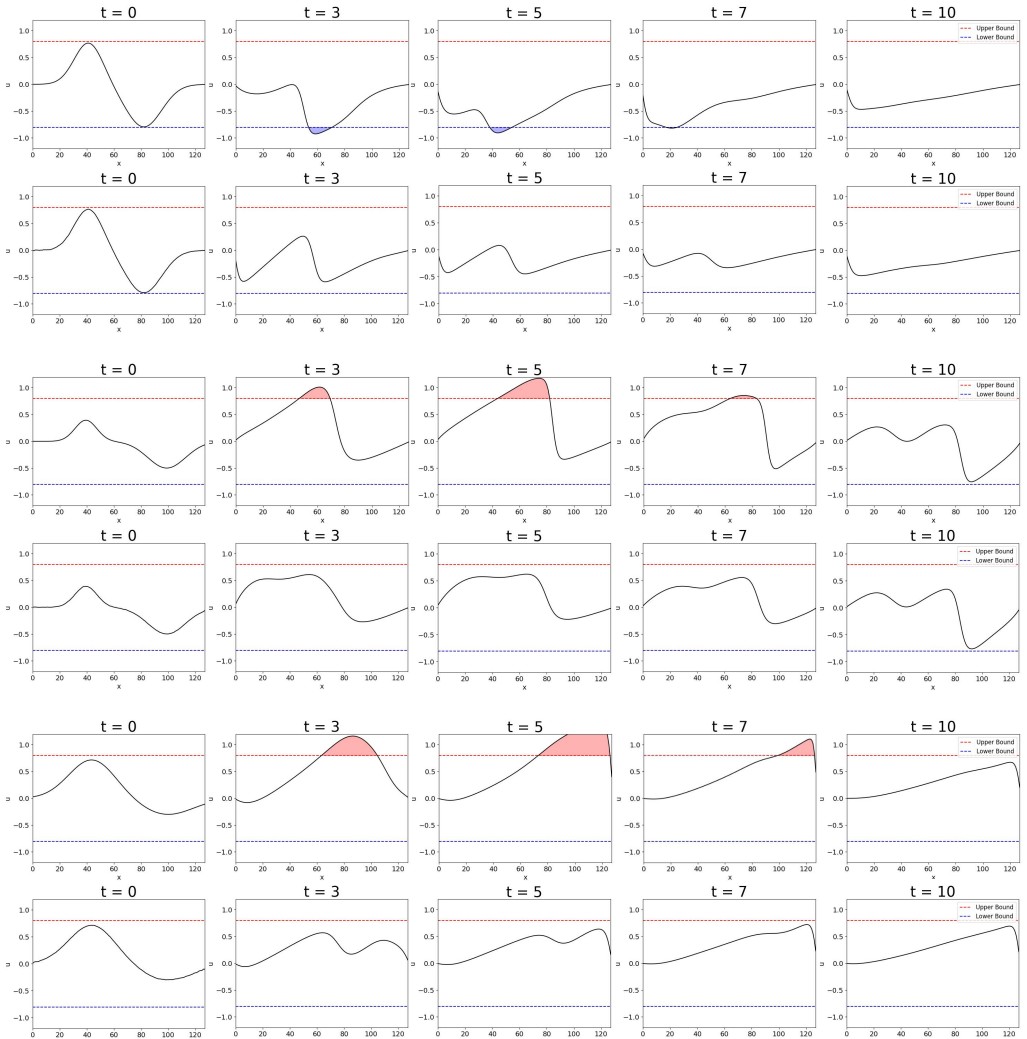

Figure 4: **Visualization of the 1D Burgers' equation.**

## A.2   2D INCOMPRESSIBLE FLUID

Here, we provide additional visualizations of the control problems of 2D incompressible fluid. From the figures, we can observe that SafeConPhy can successfully control the smoke to avoid the red hazardous region and reach the target bucket as well.

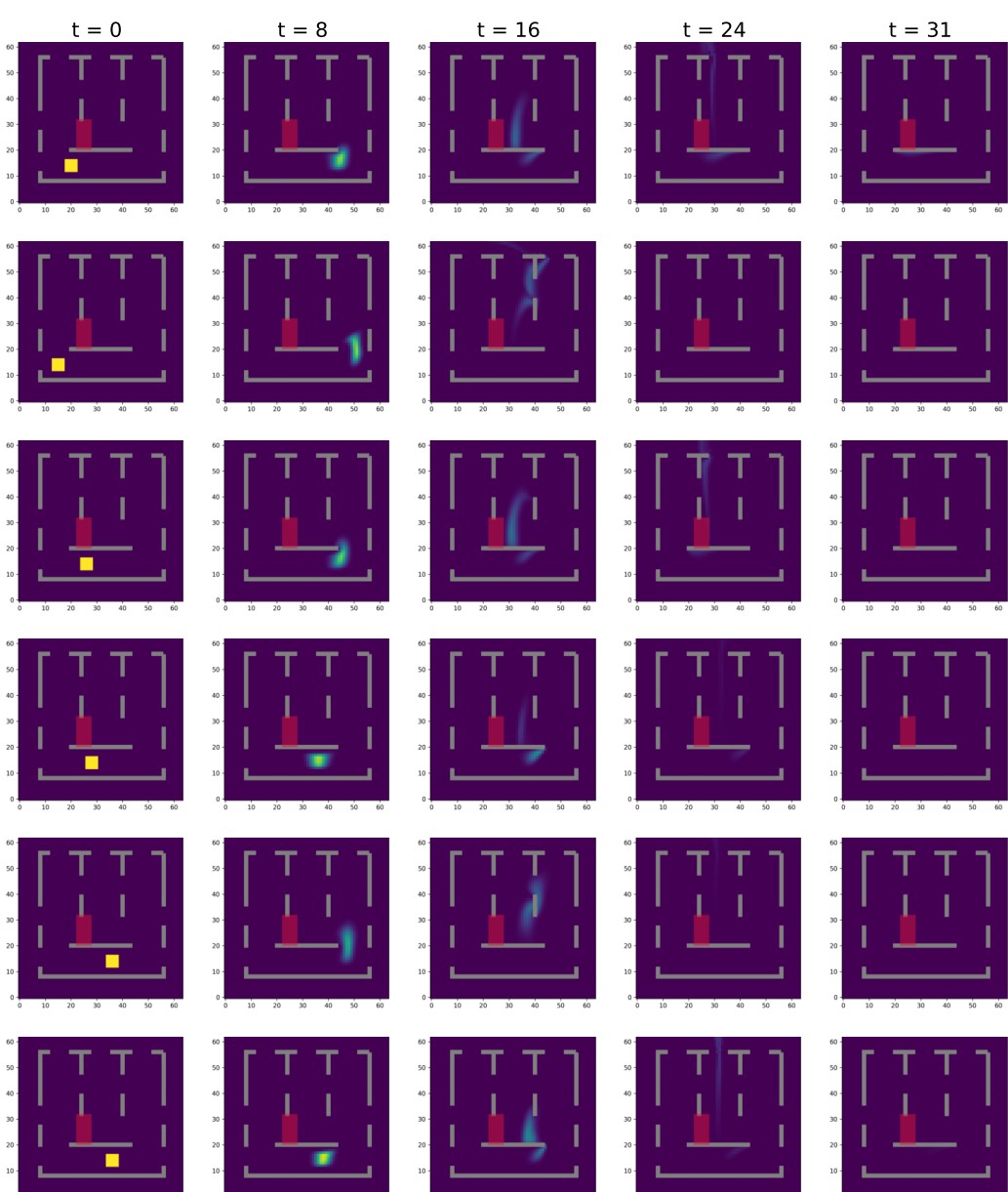

Figure 5: **Visualizations of the 2D incompressible fluid control problem.**.

## B ADDITIONAL DETAILS FOR CONFORMAL PREDICTION

Conformal prediction is a flexible framework that provides prediction intervals with guaranteed coverage probabilities for new, unseen data points, under the assumption that the data are exchangeable.

**Theoretical Foundations** The exchangeability assumption is a cornerstone of conformal prediction. It requires that the order of the data points does not affect their joint distribution, meaning that any permutation of the indices yields an identical distribution. In particular, exchangeability holds for independent and identically distributed (i.i.d.) samples, a common assumption in machine learning tasks. Ensuring exchangeability guarantees the validity of the prediction intervals constructed using conformal prediction.

**Implementation Details** To implement conformal prediction, the dataset is first split into two sub-sets: a proper training set $(Tr)$ and a calibration set $(Cal)$. A predictive model $\mu_\theta$ is trained on the training set using a specified learning algorithm $\mathcal{A}$. Once trained, the model generates predictions for the calibration set. These predictions are used to compute 'conformity scores', which measure the model's accuracy for each calibration point. Specifically, for each instance $i$ in the calibration set, the conformity score $S_i$ is defined as:

$$S_i = |\mu_\theta(X_i) - Y_i|, \quad i \in Cal.$$

Additionally, a worst-case score of $\infty$ is included to account for extreme scenarios.

Then $1 - \alpha$ quantile $q_{1-\alpha}(S)$ of the set of conformity scores is calculated, where $\alpha$ represents the desired significance level (e.g., $\alpha = 0.05$ for 95% confidence interval).

Given a new data point $X_{n+1}$, the prediction interval for its corresponding output is calculated as:

$$\hat{C}_\alpha(X_{n+1}) = [\mu_\theta(X_{n+1}) - q_{1-\alpha}(S), \mu_\theta(X_{n+1}) + q_{1-\alpha}(S)].$$

This interval provides an estimate for the range within which the true value $Y_{n+1}$ is expected to lie, with a coverage probability of at least $1 - \alpha$. Thus, conformal prediction offers a flexible and robust method for constructing prediction intervals that account for both the model's accuracy and the variability in the data.

**Theoretical Guarantees** Conformal prediction provides theoretical guarantees for finite samples (Vovk et al., 2005b; Lei et al., 2016). Specifically, for any new data point, the prediction interval satisfies the following probabilistic bound:

$$P(Y_{n+1} \in \hat{C}_\alpha(X_{n+1})) \geq 1 - \alpha.$$

This ensures that the true label $Y_{n+1}$ will fall within the predicted interval at least $1 - \alpha$ percent of the time. This framework, based on the assumption of exchangeability, provides a robust method for generating reliable prediction intervals, even in settings with limited sample sizes.

## C  ADDITIONAL DETAILS FOR 1D EXPERIMENT

### C.1  EXPERIMENT SETTING

Following the previous works (Holl et al., 2020; Wei et al., 2024), we generate the 1D Burgers' equation dataset. During inference, alongside the control sequence $\mathbf{w}(t, x)$, our diffusion model generates states $\mathbf{u}(t, x)$. Our reported evaluation metric $\mathcal{J}$ is always computed by feeding the control $\mathbf{w}(t, x)$ into the ground truth numerical solver to get $\mathbf{u}_{\text{g.t.}}(t, x)$ and computed following Eq. (17). More importantly, we consider the safety constraint and define the safety score as $\mathbf{u}^2$. In our experiment, the safety bound is fixed at 0.64, and the details of the 1D Burgers' equation dataset for safe physical system control problem are listed in Table 5.

Table 5: **Details of 1D Burgers' equation dataset.**

|  | Training Set | Calibration Set | Test Set |
|---|---|---|---|
| Unsafe Trajectories | 34,985 | 900 | 50 |
| Safe Trajectories | 4,015 | 100 | 0 |

### C.2  MODEL

The model architecture in this experiment follows the Denoising Diffusion Probabilistic Model (DDPM) (Ho et al., 2020). For control tasks, we condition on $u_0, u_T$ and apply guidance to generate the full trajectories of $u_{[0,T]}$, $f_{[0,T]}$ and the safety score $s$. The hyperparameters for the 2D-Unet architecture are recorded in Table 6.

Table 6: **Hyperparameters of 2D-Unet architecture in 1D experiment.**

| Hyperparameter Name | Value |
|---|---|
| Initial dimension | 128 |
| Convolution kernel size | 3 |
| Dimension multiplier | [1,2,4,8] |
| Resnet block groups | 1 |
| Attention hidden dimension | 32 |
| Attention heads | 4 |
| Number of training steps | 200000 |
| DDIM sampling iterations | 100 |
| $\eta$ of DDIM sampling | 1 |

## D  ADDITIONAL DETAILS FOR 2D EXPERIMENT

### D.1  EXPERIMENT SETTING

Following works from Holl et al. (2020) and Wei et al. (2024), we use the package `PhiFlow` to generate the 2D incompressible fluid dataset. The control objective and data generation is the same as before (Wei et al., 2024). The main difference between our data and previous ones is that we consider the safety constraint here. We define the safety score as the percentage of smoke passing through a specific region. This reflects the need to limit the amount of pollutants passing through certain areas in real-world scenarios, such as in a watershed.

We simulate the fluid on a $128\times128$ grid. The selected hazardous region is $[44, 36] \times [40, 64]$. Since the optimal path for smoke, starting from a left-biased position, is likely to pass through this hazardous region, this poses a greater challenge for the algorithm: how to balance safety and achieving a more optimal objective, making this a more difficult problem.

### D.2  MODEL

In this paper, the design of the three-dimensional U-net we use is based on the previous work (Ho et al., 2022). In our experiment, we utilize spatio-temporal 3D convolutions. The U-net consists of three key components: a downsampling encoder, a central module, and an upsampling decoder.

The diffusion model conditions on the initial density and uses guidance as previous mentioned to to generate the full trajectories of density, velocity, control, the objective $\mathcal{J}$ and the safety score. As $\mathcal{J}$ and $s$ are scalers, we repeat them to match other channels. The hyperparameters for the 3D U-net architecture are listed in Table 7.

## E  BASELINES

### E.1  CDT

Constraints Decision Transformer (CDT) (Liu et al., 2023b) models control as a multi-task regression problem, extending the Decision Transformer (DT) (Chen et al., 2021). It sequentially predicts returns-to-go, costs-to-go, observations, and actions, making actions dependent on previous returns and costs. The authors propose two techniques to adapt the model for safety-constrained scenarios:

1. **Stochastic Policy with Entropy Regularization**: This technique aims to reduce the risk of constraint violations due to out-of-distribution actions. In a deterministic policy, the model selects a single action based on its learned policy, which may result in unsafe actions when faced with states not well represented in the training data. By using a stochastic policy, the model samples actions from a distribution, encouraging the exploration of a wider action space. Entropy regularization further enforces diversity in the sampled actions, making the model more robust in uncertain or underrepresented situations. This approach reduces the

Table 7: **Hyperparameters of 3D-Unet architecture in 2D experiments**.

| Hyperparameter Name | Value |
|---|---|
| Number of attention heads | 4 |
| Kernel size of conv3d | (3, 3, 3) |
| Padding of conv3d | (1,1,1) |
| Stride of conv3d | (1,1,1) |
| Kernel size of downsampling | (1, 4, 4) |
| Padding of downsampling | (1, 2, 2) |
| Stride of downsampling | (0, 1, 1) |
| Kernel size of upsampling | (1, 4, 4) |
| Padding of upsampling | (1, 2, 2) |
| Stride of upsampling | (0, 1, 1) |
| Number of training steps | 200000 |
| DDIM sampling iterations | 100 |
| $\eta$ of DDIM Sampling | 1 |
| Intensity of guidance in control | 100 |
| Weight of safety term in guidance | 10000 |

likelihood of selecting unsafe actions when faced with states outside the distribution of the training set.

2. **Pareto-Frontier-Based Data Augmentation**: The technique tries to resolve the conflict between maximizing returns and adhering to safety constraints by leveraging a Pareto-frontier of the training data. The Pareto-frontier consists of trajectories that provide the highest possible return under specific safety constraints. Fitting a polynomial to the Pareto-frontier helps identify conflicting high-return and safety constraint pairs, which are then used for augmentation. The augmentation generates synthetic trajectories by relabeling safe trajectories from the Pareto-frontier with higher returns and assigning higher or equal safety constraints. This encourages the model to imitate the most rewarding, safe trajectories when the desired return given the safety constraint is infeasible.

In the 2D incompressible fluid experiment, the model fails to extrapolate safe trajectories with higher returns due to the training data **covering a broad range of costs, while the desired cost lies in a narrow range**. The augmentation treats all safe trajectories on the Pareto-frontier equally, without emphasizing the region of interest. To investigate, we filtered the training data to include only safe trajectories under the desired safety bound and retrained the model, showing that the data complexity exceeds CDT's capacity. This complexity, however, could potentially be addressed by SafeConPhy, which is indicated in Table 3. We use the official CDT implementation and follow DT guidelines to sweep desired returns and cost constraints in testing time. In the 1D Burgers experiment, we modify the control objective from the original mean squared error $\mathcal{J}$ between the prediction and target, to an exponential form $\exp(-\mathcal{J})$. This new objective is bounded within $[0, 1]$, which better aligns with the reward-maximizing setup in reinforcement learning used by CDT. For the 2D incompressible fluid setup, the state, action, and cost prediction heads each consist of 3-layer MLPs with the transformer's hidden dimension as the inner size.

### E.2  BC-ALL

The Behavior Cloning (BC) algorithm, introduced by (Pomerleau, 1988), is a foundational technique in imitation learning. BC is designed to derive policies directly from expert demonstrations, utilizing supervised learning to associate states with corresponding actions. This method eliminates the necessity for exploratory steps commonly required in reinforcement learning by replicating the actions observed in expert demonstrations. One of the significant advantages of BC is that it does not involve interacting with the environment during the training phase, which streamlines the learning process and diminishes the demand for computational resources.

In this approach, a policy network is trained using standard supervised learning strategies aimed at reducing the discrepancy between the actions predicted by the model and those performed by the

Table 8: **Hyperparameters of 1D CDT**.

| 1D Burgers' | All data | Safe filtered |
|---|---|---|
| State Dimension | 256 | 256 |
| Action Dimension | 128 | 128 |
| Hidden Dimension | 1024 | 1024 |
| Number of Transformer Blocks | 2 | 2 |
| Number of Attention Heads | 8 | 8 |
| Horizon (Sequence Length) | 5 | 5 |
| Learning Rate | 1e-4 | 1e-4 |
| Batch Size | 64 | 64 |
| Weight Decay | 1e-5 | 1e-5 |
| Learning Steps | 1,000,000 | 1,000,000 |
| Learning Rate Warmup Steps | 500 | 500 |
| Pareto-Frontier Fitted Polynomial Degree | 0 | 4 |
| Augmentation Data Percentage | 0.3 | 0.3 |
| Max Augment Reward | 10.0 | 10.0 |
| Min Augment Reward | 1.0 | 1.0 |
| Target Entropy | -128 | -128 |
| Testing Time Sweep Returns | 9.0, 9.9 | 9.0, 9.9 |
| Testing Time Sweep Costs | 0.0, 1.0, 2.0, 3.0 | 0.0, 1.0, 2.0, 3.0 |

Table 9: **Hyperparameters of 2D CDT**.

| 2D incompressible fluid | All data | Safe filtered |
|---|---|---|
| State Dimension | $3\times64\times64$ | $3\times64\times64$ |
| Action Dimension | $2\times64\times64$ | $2\times64\times64$ |
| Hidden Dimension | 512 | 512 |
| Number of Transformer Blocks | 3 | 3 |
| Number of Attention Heads | 8 | 8 |
| Horizon (Sequence Length) | 10 | 10 |
| Learning Rate | 1e-4 | 1e-4 |
| Batch Size | 8 | 8 |
| Weight Decay | 1e-5 | 1e-5 |
| Learning Steps | 1,000,000 | 1,000,000 |
| Learning Rate Warmup Steps | 500 | 500 |
| Pareto-Frontier Fitted Polynomial Degree | 4 | 4 |
| Augmentation Data Percentage | 0.3 | 0.3 |
| Max Augment Reward | 32.0 | 32.0 |
| Min Augment Reward | 1.0 | 1.0 |
| Target Entropy | $-(2\times64\times64)$ | $-(2\times64\times64)$ |
| Testing Time Sweep Returns | 18.0, 32.0 | 18.0, 32.0 |
| Testing Time Sweep Costs | 0.0, 0.1, 0.2 | 0.0, 0.1, 0.2 |

expert in the dataset. The commonly used loss function for this purpose is the mean squared error between the predicted actions and expert actions. The dataset for training comprises state-action pairs harvested from these expert demonstrations. In the work, we employ the implementation as Liu et al. (2023a).

### E.3  BC-SAFE

Following the baseline in Liu et al. (2023a), the BC-Safe is fed with only safe trajectories filtered from the training dataset, satisfies most safety requirements, although with conservative performance and lower rewards. Others are same as BC-All, except for the safe trajectories.

### E.4 SL-LAG

Hwang et al. (2022) introduces a supervised learning (SL) based method to control PDE systems. It first trains a neural surrogate model to capture the PDE dynamics, which includes a VAE to compress PDE states and controls into the latent space and another model to learn the PDE's time evolution in the latent space. To obtain the optimal control sequence, SL can compute the gradient $\nabla_{\mathbf{w}}\mathcal{J}$, where $\mathcal{J}$ is the control objective and $f$ is the input control sequence. Then iterative gradient optimization can be executed to improve the control sequence.

To ensure that the optimal control is compatible with the hard constraint in our experiments, we follow Chow et al. (2018) to apply the Lagrange optimization method to the constrained optimization. Specifically, we iteratively solve the optimization problem below:

$$\max_{\lambda \geq 0} \min_{\mathbf{w}} \mathcal{J}(\mathbf{w}) + \lambda(s(\mathbf{u}(\mathbf{w})) - c). \tag{21}$$

We denote the modified SL method SL-Lag.

Table 10: **Hyperparameters of network architecture and training for SL-Lag in 1D Burgers' experiment**.

| Hyperparameter name | Value |
|---|---|
| Initialization value of $\mathbf{w}$ | 0.001 |
| Optimizer of $\mathbf{w}$ | LBFGS |
| Learning rate of $\mathbf{w}$ | 0.1 |
| Initialization value of the Lagrange multiplier $\lambda$ | 0 |
| Optimizer of $\mathbf{w}$ | plain GD |
| Learning rate of $\lambda$ | 10 |
| Iteration of $\lambda$ | 2 |
| Loss function | MSE |

### E.5 TREBI

In Lin et al. (2023), the diffusion model is adopted for the planning task under safety budgets. It generates trajectory under safety constraints using classifier-guidance Dhariwal & Nichol (2021) by adding a safety loss to the reward guidance following Diffuser Janner et al. (2022).

However, the original setting is different from our experiments. In our 1D Burgers' equation control, our objective is that a certain state equals the target state, which in-painting diffusion condition Janner et al. (2022) is more appropriate for. Furthermore, as in our method and in (Wei et al., 2024), a conditional diffusion model can be learned to tackle the objective more directly. In addition, TREBI follows the setting in Diffuser where the interaction with the environment is allowed which in our experiments becomes an MPC method. Note that the reported results of our method do not involve interaction with the surrogate model (though our method can easily adapted to be an MPC method). Thus, the results of TREBI in Table 2 and 3 have an unfair advantage.

Therefore, for 1D Burgers' experiment, we conducted different experiments on TREBI including (1) planning multiple times with interaction with the surrogate model or (2) planning only once, and with target state conditioning or target state guidance. The target state guidance + planning multiple times turned out the best and is reported in Table 2. For 2D smoke control, the target is not a state constraint but a reward, and the planning multiple-step setting is too computationally expensive. To this end, we use reward guidance with planning one single time, which is identical to the ablation study of our method in Table 4. The hyperparameters of the 1D experiment are reported in Table 11, and those of 2D are the same as reported before.

### E.6 PID

Propercentageal Integral Derivative (PID) control (Li et al., 2006) is a versatile and effective method widely employed in numerous control scenarios. For 1D control task, we mainly implement the PID baseline adapted from Wei et al. (2024). More detailed configurations can be found in Table 12

Table 11: **Hyperparameters of network architecture and training for TREBI in 1D Burgers' experiment**.

| Hyperparameter name | Value |
|---|---|
| Reward guidance intensity | 50 |
| Safety guidance intensity | 1 |
| Cost budget | 0.64 |
| Number of guidance steps | 10 |
| Denoising steps | 200 |
| Sampling algorithm | DDPM |
| U-Net dimension | 64 |
| U-Net dimension mltiplications | 1, 2, 4, 8 |
| Planning horizon | 8 steps |
| Optimizer | Adam |
| Learning rate | 0.0002 |
| Batch size | 16 |
| Loss function | MSE |

Table 12: **Hyperparameters of network architecture and training for ANN PID**.

| Hyperparameter name | Value |
|---|---|
| Kernel size of conv1d | 3 |
| Padding of conv1d | 1 |
| Stride of conv1d | 1 |
| Activation function | Softsign |
| Batch size | 16 |
| Optimizer | Adam |
| Learning rate | 0.0001 |
| Loss function | MAE |

