# OpenReview forum: "A Probabilistic Generative Method for Safe Physical System Control Problems"
_ICLR.cc/2025/Conference — Submitted to ICLR 2025_

### Official Review · Reviewer_88cy · 2024-10-17

**Soundness:** 1
**Presentation:** 2
**Contribution:** 1
**Rating:** 3
**Confidence:** 5

**Summary:**

This paper introduces a probabilistic generative method for safe control problems of complex physical systems and provides a provable probabilistic estimate of the safety score’s upper bound. Experiment results on 1D Burgers’ equation and 2D incompressible fluid demonstrate that the proposed SafeConPhy can achieve a lower control objective based on satisfying safety constraints.

**Strengths:**

This paper introduces a probabilistic generative method for safe control problems of complex physical systems and provides a provable probabilistic estimate of the safety score’s upper bound.

**Weaknesses:**

1. The reviewer cannot agree with many critical statements in the paper. For example, the ``In contrast, while safe reinforcement learning methods consider safety, they typically fail to provide guarantees for satisfying safety constraints.” Safe RL has been well studied recently and generated critical and solid results about safety, which the paper ignores. For example, the work in [1, 2] presents verifiable stability and safety, and the reference [3] includes more related work. The paper is expected to be rich in its literature review and comparisons.

2. The paper has a critical underlying assumption: the PDEs of the application systems are precisely known. This assumption is impractical because, for our real systems, such as power systems, vehicles, drones, robotics, etc., their real dynamics are very complex, and we only have partial knowledge about the PDEs.  Because of the underlying assumption, I think the proposed framework may face many issues in its real implementation. So, I suggest the paper can identify the potential problems and present the discussions.

3. The proposed framework builds on the DiffPhyCon. For example, the significant technical Eqs. (2)-(4) and Eqs. (6)-(12) are directly cited from the existing work in (Wei et al., 2024) and (Tibshirani et al., 2019). Lemma 1 is a theoretical contribution. I cannot conclude if it is the paper's contribution or cited from others because it lacks details. Meanwhile, the paper does not include implementation in real systems, so I review this paper as a theory paper, and my conclusion is that the paper does not have sufficient theoretical contributions.

4. We have typical approaches to achieving the same safe control tasks: ML-based controllers and traditional controllers in the control community. The paper lacks comparisons with conventional controllers (e.g., MPC, PID, and adaptive controllers) and also safe RLs, like the ones in [1]-[3]. The paper is expected to include such comparisons to demonstrate the advantages of the proposed method.

5. All the experiment examples are numerical and toy, and there is a significant mismatch with real physical engineering systems. Therefore, I do not think they are convincing. To address the concern, the paper could consider experiments on more realistic safety-critical systems or simulators (such as autonomous vehicles and robots in Nvidia Isaac Sim).

6. Digital sensor sampling transforms our real system into a discrete-time system, further impeding the application of the proposed framework in real systems. This is because the proposed approach needs precise PDEs, while digital sampling can induce model errors that the PDEs cannot capture. The paper shall include sections to discuss the implementation of the work in the real systems and the influence of the model error on the system and safety performances.

7. The technical parts are hard to follow. For example, Eqs. (2)-(4) are directly cited from (Wei et al., 2024), and it is expected the paper can include explanations about the key parameters in Eqs. (2)-(4).

8. The literature review lacks safe DRL (deterministic and probabilistic safety) and safety-critical control, such as in [1-3].

References:

[1] Zhao, L., Gatsis, K., & Papachristodoulou, A. (2023, December). Stable and safe reinforcement learning via a barrier-Lyapunov actor-critic approach. In 2023 62nd IEEE Conference on Decision and Control, pp. 1320-1325.

[2] Cao, H., Mao, Y., Sha, L., & Caccamo, M. Physics-Regulated Deep Reinforcement Learning: Invariant Embeddings. In The Twelfth International Conference on Learning Representations, 2024.

[3] Gu, S., Yang, L., Du, Y., Chen, G., Walter, F., Wang, J., & Knoll, A. (2022). A review of safe reinforcement learning: Methods, theory and applications. arXiv preprint arXiv:2205.10330.

**Questions:**

1. Is the Lemma 1 a theoretical result of this paper? If so, I cannot find its formal proof. If not, what is its source reference?

2. Can the safety consideration degrade the system’s mission or control performance? The paper lacks such critical study and discussion.

3. What does "SafeConPhy is able to achieve a lower control objective" mean? lower control performance?

---

### Official Review · Reviewer_9vw1 · 2024-10-31

**Soundness:** 2
**Presentation:** 2
**Contribution:** 3
**Rating:** 5
**Confidence:** 4

**Summary:**

This paper considers PDE control problems with constraints, and proposes a probabilistic generative method inspired by DiffPhyCon that accounts for safety. Experiment results on 1d burgers equation and 2d incompressible fluid are conducted and performance vs. safety tradeoff improvements is shown over several baselines.

While the idea of the paper seems novel and the experiment is reproducible (code available anonymously), I have several concerns on the effectiveness of the safety guarantees and the presentation of the paper.

**Strengths:**

The idea of incorporating safety constraints into diffusion based methods for PDE control seems novel. The experiments are reproducible given the provided code.

**Weaknesses:**

The writing of the paper needs improvements.

Notation inconsistency:
In the problem formulation and 1D example, u is the state and w is the control. However, in the 2D example, boldsymbol v and non-bold symbol p are used and the external force is f. Still, there is no argument on which variables are states and which are control inputs in this case.

Several arguments are confusing, e.g.:
‘We need to minimize the control objective while satisfying physical constraints and constraining the safety score to stay below the bound, which requires a careful balance between safety and performance’
Isn't this by formulation of the optimization problem? The optimal solution to the constrained optimization should optimize performance while satisfying safety?

Abbreviations are used without definition:
OOD is used without definition in section 2.2, does it mean ‘out of distribution’?

Variables are used without definition, e.g.:
In section 3.2, what are E_\theta (u, w) and p(u, w)? Is p(u, w) here the same as the one in section 4.2 above equation (6)?

Besides, I have concerns on the theoretical guarantees and the empirical effectiveness of the proposed method. See the questions section for detail.

**Questions:**

The authors claim that the proposed method has safety guarantees, but the results indicate unsafe behaviors in the 2D example. Any explanations?

No proof for Lemma 1? I did not find it anywhere in the paper or in the appendix.

In Table 3, it seems that BC-safe has the least safety constraint violation, but the proposed method is marked bold?

Are there any experimental comparisons with DiffPhyCon since it is a highly related work?

---

### Official Review · Reviewer_FY42 · 2024-11-02

**Soundness:** 2
**Presentation:** 2
**Contribution:** 2
**Rating:** 3
**Confidence:** 4

**Summary:**

This paper studies how to provide probabilistic safety guarantees for diffusion control policy for physical systems. Specifically, conformal prediction (CP) is used to provide such safety guarantees. To handle distribution shift between calibration and test datasets, the authors use weighted CP to be aware of covariate shift. The proposed algorithm is evaluated on a 1D system and a 2D system and outperforms some baselines. Ablation study on fine-tuning is also conducted.

**Strengths:**

1. I think this paper is overall clearly presented. The studied safety-critical control problem is important and relevant to ICLR community.
2. The experimental results look promising.

**Weaknesses:**

1. It looks to me that the comparison with related work is incomplete, which makes me concerned on the contribution of this work. I'd recommend two relevant papers to the authors: [1] and [2].

  In [1], the conformal prediction (CP) is also utilized to ensure the safety/robustness of the diffusion model and the approach is applied in more complex physical systems such as high-dimensional robotic systems. Particularly, an uncertainty-related loss from CP is proposed in [1] and I think it is similar to L_safe, which is proposed by the authors in (15). I strongly recommend that the authors compare the current submission with [1] regarding its similarities and difference. Also, I suggest the authors to clarify the novelty of this paper compared with [1].

In [2], a Lyapunov-based safety shield is used to ensure the safety of diffusion model and it has no probabilistic safety risk. In this sense, CP guarantee is weaker than [2]. I'd suggest the authors to compare with [2] as well regarding its strengths and weaknesses.

2. The authors claim that the proposed approach can be applied to *complex* physical systems. However, only 1D and 2D systems have been demonstrated in experiments. In the literature like [1-3], diffusion policy has been shown in various complex and high-dimensional robotic systems including real-world robotic arm systems. Therefore, in my opinion, if the authors stress their focus on safety for *complex* physical systems, then experiments on complex and challenging robotic tasks are necessary.

3. I think there might be a technical problem in Section 4.2. Specifically, the generated data distribution (p_theta) is generally not the same as the training/calibration data distribution (p), so the jump from (8) to (9) is not valid unless the authors make the assumption that p_theta is the same as p, which is a strong assumption though because it really depends on the training quality of the diffusion model. This also influences the validity of Lemma 1.

Even if the diffusion training quality is high, the assumption still does not hold. Because the authors modified the vanilla diffusion training loss by adding a safe loss, which can change the diffusion generation distribution and make it deviate from the original training data. That is to say, the data recovered/generated by the diffusion model will not be from the training data distribution due to the new loss term. Thus, I don't think it is reasonable to assume that p_theta and p are the same.

I include more concerns and questions in the Questions section.

[1] Conformal Prediction for Uncertainty-Aware Planning with Diffusion Dynamics Model. Advances in Neural Information Processing Systems 2024.

[2] Safediffuser: Safe planning with diffusion probabilistic models. arXiv preprint arXiv:2306.00148.

[3] Diffusion policy: Visuomotor policy learning via action diffusion. The International Journal of Robotics Research 2023.

**Questions:**

1. I am not sure if the authors are considering the model-free case or model-based setting? I would suggest the authors to clarify this issue. If the model-based case is considered in this paper, then the authors should also compare with more classical control methods such as model predictive control [4] and optimal safe control [5] instead of just model-free PID. In addition, in the presented two experiments (1D and 2D systems), optimal safe control approaches such as [5] may be able handle it if the model is known.

2. I am not fully following "With the influence of G ..." (the equation in Line 293 and 294) and would appreciate it if the authors can make some clarifications.

3. I suggest the authors should also compare with [1] and [2] in experiments, which are two safety-concerned diffusion policy approaches.

[4] Model predictive control: Theory and practice—A survey. Automatica 1989.

[5] A time-dependent Hamilton-Jacobi formulation of reachable sets for continuous dynamic games. IEEE Transactions on automatic control 2005.

---

### Official Review · Reviewer_xBBv · 2024-11-03

**Soundness:** 2
**Presentation:** 1
**Contribution:** 2
**Rating:** 5
**Confidence:** 3

**Summary:**

This paper studies the problem of controlling a PDE under safety constraints. Building on prior work applying diffusion models to PDE control, the authors train a diffusion model to learn the joint distribution over PDE state and optimal control input (i.e. jointly learning the PDE dynamics and the control law). They then apply weighted conformal prediction to estimate the worst-case safety violation that could be observed at a specified risk level, then fine-tune the joint control-dynamics diffusion model to reduce this worst case. The authors evaluate on 1D and 2D test cases and compare against a large number of baseline methods.

**Strengths:**

The paper studies an interesting problem (safe control of PDE dynamics) and compares with a large list of credible baselines. The figures do a good job of illustrating both the approach (Fig. 1) and the results (Figs. 2 and 3).

**Weaknesses:**

The structure of the paper is difficult to follow (e.g. not specifying that the focus is on PDE control until the third page), and a number of key details are not explained (see questions below). Additionally, empirical results comparing the claimed upper bound with the observed upper bound are not included, preventing readers from assessing whether the proposed method is "certifiable" as the authors claim.

I also have a concern about the methodology. It would seem that the proposed fine tuning step risks changing the learned PDE model to make it easier to  control rather than solely changing the control law. What steps could you take to make sure that fine tuning only changes the control law and not the learned dynamics? This seems like a critical issue to me, because the derivation of the upper bound on page 6 relies on the assumption that the diffusion model exactly learns the true dynamics (i.e. that $p_\theta(u_i, w_i) \approx p(u_i, w_i)$). If the fine tuning step changes $p_\theta$ to minimize $s_+$, does this assumption still hold?

**Summary of my review:** I think this paper has the potential to be a strong paper, but it is not ready for acceptance in its current form. If the authors could revise to make the paper more clear, as well as address the concerns raised here, I would consider increasing my score.

Minor issues
- PDE control is not mentioned in the abstract or introduction until the third page. If this is the focus of the paper, say so early on.
- It is misleading to bold the entire row for SafeConPhy in Table 2. Standard practice is to bold the entry in each column that achieves the lowest score. So $s_{norm}$ should have PID bolded, and all methods that achieve 0% safety violations should be bolded in those columns.
- Line 431: is there a typo in the definitions of $\mathcal{N}_1$ and $\mathcal{N}_2$? Should it be $s(u_i)$ rather than $u_i$?
- The "Limitation and Future Work" section is mainly focused on future work; I would like to see a more thorough discussion of the limitations of this work (e.g. what are some of the barriers that prevented experiments on hardware?).

**Questions:**

1. Can you define "complex physical system"? This is not a precise technical term.
2.  What data is contained in the training and calibration datasets? Control/state pairs? Time series?
3. Are you assuming that the true dynamics are deterministic? That seems to be the assumption behind simplifying equation 7
4. You assume that $dp_\theta(u_i, w_i) \approx dp(u_i, w_i)$ in Eq. 8. Do you have empirical evidence for this assumption? This would imply that the diffusion model has learned the calibration data distribution perfectly.
5. Your method learns a joint distribution over both state and control, implicitly learning the dynamics that relate the two. Is there a risk that fine-tuning the model just makes the implicit dynamics less likely to become unsafe (even if that means they are different from the real dynamics) rather than changing the learned control law? How would you mitigate this risk?
6. Can you confirm that the evaluation metrics in Tables 2 and 3 are calculated using the true dynamics and not the learned dynamics (given my question 5 above)? Looking at your code in `inference.py`, it seems that it may be calculating safety scores using the state trajectory generated by the learned model rather than the true PDE, which would not be an accurate evaluation. I can see where your evaluation for PID uses `burgers_numeric_solve`, but I do not see this function used in your evaluation for SafeConPhy. I am likely missing something in your code, so if you could explain how you are evaluating the safety rate (and point to the relevant part of the code), I would greatly appreciate it.
7. How does the empirically observed safety score compare to the upper bound found via conformal prediction? How tight/loose is this bound?
8. In table 3, methods that have $s_{norm} < 1$ (indicating that no samples were unsafe) also have $\text{max}(s-s_0, 0) > 0$, implying that at least one sample exceeds the safety bound. This would appear to be a contradiction, but perhaps I am not understanding how you have defined these metrics. Could you please explain?

---

### Meta-Review · Area_Chair_mrmX · 2024-12-20

**Metareview:**

This paper introduces a probabilistic generative method for controlling a PDE under safety constraints. The reviewers raised many issues with the paper and suggested multiple improvements. The authors did not respond during the rebuttal period. There is a clear consensus that this paper does not meet the bar for acceptance.

**Additional Comments On Reviewer Discussion:**

The authors did not respond to the reviewers.

---

### Decision · Program_Chairs · 2025-01-22

Reject